# *Nelumbo nucifera* Receptaculum Extract Suppresses Angiotensin II-Induced Cardiomyocyte Hypertrophy

**DOI:** 10.3390/molecules24091647

**Published:** 2019-04-26

**Authors:** Soyoung Cho, Hyun Woo Cho, Kyeong Wan Woo, Jisu Jeong, Juyeon Lim, Sungha Park, Miran Seo, Soyeon Lim

**Affiliations:** 1Graduate Program in Science for Aging, Yonsei University, Seoul 120-752, Korea; sange119@naver.com (S.C.); jisu2082@yuhs.ac (J.J.); lovelynar@yuhs.ac (J.L.); 2Korean Medicinal Herbs Research Team, National Development Institute of Korean Medicine, 288 Udeuraendeu-gil, Jeollanam-do 59337, Korea; johw7@nikom.or.kr (H.W.C.); ddoksory@nikom.or.kr (K.W.W.); 3Cardiovascular Research Institute, Division of Cardiology, Yonsei University College of Medicine, Seoul 120-752, Korea; shpark0530@yuhs.ac; 4Severance Integrative Research Institute for Cerebral & Cardiovascular Diseases, Yonsei University College of Medicine, Seoul 120-752, Korea; 5Institute for Bio-Medical Convergence, College of Medicine, Catholic Kwandong University, Gangneung, Gangwon-do 25601, Korea

**Keywords:** *Nelumbo nucifera* receptaculum, angiotensin II, cardiomyocyte hypertrophy

## Abstract

*Nelumbo nucifera* Gaertn. (lotus) is an important medicinal plant, and many parts of the plant have been investigated for their therapeutic effects. However, the therapeutic effect of receptacles of lotuses on pathological cardiomyocyte hypertrophy has not been investigated yet. Therefore, the current study aimed to determine the protective effect of lotus against angiotensin II (Ang II)-induced cardiomyocyte hypertrophy in vitro. Ang II was used to induce hypertrophy of H9c2 cells. The lotus receptacle powder (MeOH extract of receptaculum Nelumbinis; MRN) used in the experiments was prepared by MeOH extraction and subsequent evaporation. To evaluate the effect of MRN on cardiomyocyte hypertrophy, cell size, protein synthesis, and hypertrophic marker expressions were examined. The antioxidant ability of MRN was determined by using CM-H_2_DCFDA, a general oxidative stress indicator. Ang II-induced cardiomyocyte hypertrophy was significantly attenuated by 5 µg/mL of MRN, as confirmed by the reductions in cell size, protein synthesis, and hypertrophic marker expression. MRN also attenuated Ang II-induced excessive intracellular reactive oxygen species (ROS) production through the suppression of protein kinase C (PKC), extracellular-signal-regulated kinase (ERK), and NF-κB activation and subsequent type I angiotensin receptor (AT1R), receptor for advanced glycation end products (RAGE), and NADPH oxidase (NOX) expression. MRN exerted a significant protective effect against Ang II-induced cardiomyocyte hypertrophy through suppression of PKC–ERK signaling, and this subsequently led to attenuation of intracellular ROS production.

## 1. Introduction

Pathological cardiac hypertrophy is an important and independent risk factor for the development of heart failure, one of the leading causes of death worldwide [1,2]. Pathological cardiac hypertrophy has various risk factors, such as chronic or abnormal hemodynamic stress, myocardial injury, and excessive neurohumoral activation [2]. Its characteristics include an increase in cardiomyocyte size, fetal gene activation, and interstitial fibrosis, which result in depressed cardiac function that is usually irreversible [3]. Angiotensin II (Ang II), an important regulator of the renin–angiotensin system, plays a major role in hypertension, which results in cardiac hypertrophy through the activation of type I angiotensin receptor (AT1R) [4]. AT1R is expressed in various tissues, including the heart, blood vessels, and the brain [5], and the importance of AT1R-mediated signaling in cardiac hypertrophy has been demonstrated through the use of AT1R blockers, knockout mice, and inhibitors [6,7,8]. In cardiomyocytes, activated AT1R subsequently activates the Gq protein, which causes protein kinase C (PKC) activation and intracellular calcium release, resulting in mitogen-activated protein kinase (MAPK) activation [9]. The activation of MAPK and the subsequent activation of extracellular-signal-regulated kinase (ERK) are the critical signaling pathways associated with cardiac hypertrophy and heart failure that occur in response to a variety of external stimuli, such as Ang II and noradrenaline [10]. Ang II is also known to produce reactive oxygen species (ROS) via NADPH oxidases (NOXs), which are activated by PKC or intracellular calcium, and this excessive cytosolic ROS can stimulate mitochondrial ROS overproduction. Again, the increased mitochondrial ROS can exit the mitochondria and provide the feed-forward stimulation of cytoplasmic NOX and finally lead to cardiac hypertrophy [11,12]. 

*Nelumbo nucifera* Gaertn. (common name, lotus), one of two species of the family Nymphaeaceae, has been cultivated in several countries, including Egypt, China, India, Japan, and South Korea, and has been used for over 2000 years as a traditional medicine and dietary component [13]. Lotus has become one of the most important medicinal plants, and different parts of it, such as the flower, leaf, seed, and rhizome, have been investigated for their therapeutic effects [14]. Lotus extracts are reported to be effective for several diseases, including cancers, cardiac diseases, and hepatic disease [15,16,17]. With regard to cardiovascular diseases, although the therapeutic effects of lotus depend on the extraction methods or the parts used for extraction, lotus generally appears to have anti-inflammatory, antioxidant, and anti-obesity effects. For example, receptaculum Nelumbinis (receptacle of *Nelumbo nucifera* Gaertn.), of which the major components are hyperoside, isoquercitrin, quercetin, isorhamnetin, and syringetin, has been reported to have a strong antioxidant activity [13]. However, unlike the other parts of lotus, only little information is available on the effect of receptaculum Nelumbinis in cardiac diseases, especially cardiac hypertrophy. As excessive oxidative stress is a major cause of cardiac disease and previous reports have shown the antioxidant activities of receptaculum Nelumbinis extract, we hypothesized that a MeOH extract of receptaculum Nelumbinis (MRN) may exert a protective effect on pathological cardiomyocyte hypertrophy, and therefore examined the effect of MRN on intracellular ROS production and the related signaling pathways.

## 2. Results

### 2.1. MRN Attenuated Ang II-Induced Cardiomyocyte Hypertrophy

To determine the optimal concentration of MRN to treat H9c2, 1–50 µg/mL MRN was incubated in the presence or absence of Ang II for 48 h. Cells treated with up to 20 µg/mL MRN exhibited no significant reduction in survival compared with the control cells (Figure 1A), and LDH activity in the culture medium was significantly increased by 10 µg/mL MRN treatment compared with the control in the presence or absence of Ang II treatment (Figure 1B). We then investigated the anti-hypertrophic effect of 1–10 µg/mL MRN in Ang II-induced cardiomyocyte hypertrophy by using H9c2 cells. Ang II-induced cardiomyocyte hypertrophy was confirmed through the measurement of various indices, including an increase in cell size, protein synthesis, and expression of hypertrophic markers. Ang II stimulation for 48 h resulted in a significant increase in the surface area of H9c2 cells. However, this was markedly attenuated by MRN treatment from 5 µg/mL (Figure 2A,B). Treatment with 10 µg/mL MRN also exerted anti-hypertrophic effects, but significant cytotoxicity was observed. Therefore, 5 µg/mL MRN was used for further experiments. Increases in the protein expression of NFATc-1, ANP, BNP, and MLC2v that were induced by Ang II stimulation were attenuated by MRN treatment (Figure 2C,D). We also investigated protein synthesis, another critical hypertrophic phenomenon. Ang II-induced protein synthesis was strongly induced at 48 h and this was markedly attenuated by MRN treatment (Figure 2E,F). These results showed that MRN significantly suppressed Ang II-induced cardiomyocyte hypertrophy.

### 2.2. MRN Inhibited Intracellular ROS Levels in Ang II-Induced Cardiomyocyte Hypertrophy

We then investigated whether MRN inhibited intracellular ROS, a critical factor in Ang II-induced cardiac hypertrophy. Intracellular ROS levels were examined after incubation with Ang II for 30 min or 24 h, and Ang II induced a significant increase in DCFDA fluorescence. MRN treatment markedly suppressed the increase in intracellular ROS levels to those of the control (Figure 3A). The molecular mechanisms of Ang II in cardiac hypertrophy are involved with the stimulation of NOXs as the major source of ROS [11]. Given that Ang II strongly stimulates NOX expression and activation [18], we examined the protein level of NOX isoforms in the presence or absence of MRN and Ang II stimulation. Increased expression of NOX2 and NOX4 was found in Ang II-stimulated H9c2 cells, and this was attenuated by MRN treatment (Figure 3B,C). However, Ang II-induced NOX1 overexpression was not significantly changed by MRN. It is well known that NF-κB activation is critical for Ang II-mediated signaling pathways [19,20], and NOX transcription as well as NOX-dependent ROS formation are regulated by NF-κB signaling [21]. Therefore, NF-κB activation was examined, and MRN attenuated Ang II-induced NF-κB activation (Figure 3D,E).

### 2.3. MRN Modulates Cardiomyocyte Hypertrophy through Regulation of PKC—ERK Signaling Pathway

As the regulatory action of Ang II occurs mainly via AT1R activation, we investigated the protein expression of AT1R. The protein expression of AT1R after Ang II treatment was increased, and MRN treatment significantly attenuated this increase in the AT1R protein (Figure 4A,B). To further explore the mechanisms underlying the anti-hypertrophic effects of MRN in Ang II-treated H9c2 cells, receptor for advanced glycation end products (RAGE) expression was also investigated. RAGE is known to be activated via HMGB1 in AT1R activation in cardiomyocytes [22]. RAGE protein expression was upregulated under Ang II stimulation, in which MRN attenuated RAGE overexpression. The level of secreted HMGB1 was also estimated in the cell culture medium to check the ability of MRN to inhibit AT1R-mediated RAGE activation. Ang II treatment increased HMGB1 secretion compared with the normal control, and MRN significantly attenuated HMGB1 secretion (Figure 4C). The phosphorylation of PKC and ERK1/2, key mediators of the cardiac hypertrophic signaling pathway associated with AT1R–RAGE activation, were detected by immunoblotting. Ang II induced the activation of PKC and ERK1/2, and this was markedly attenuated by MRN treatment (Figure 4D,E). 

## 3. Discussion

Herbal medicines have significantly contributed to the development of modern medicines. Although synthetic drugs have shown significant therapeutic effects for various diseases, severe and unexpected side effects sometimes occur [23,24]. As herbal medicines, especially medicinal plants, have been used as alternative medicines for a long time and have been shown to have almost no side effects, they continue to be investigated for many purposes. In this study, we used receptaculum Nelumbinis, a notable herbal medicine, as the experimental material. MRN exerted a significant anti-hypertrophic effect in H9c2 cells via the inhibition of the AT1R-mediated signaling pathway, suppressing excessive intracellular ROS production. In particular, it is known that NOX-derived ROS signaling is very important for Ang II-mediated cardiovascular injury [25]. As key proteins for AT1R-mediated ROS production, PKC activation and the protein level of NOX isoforms were examined. MRN significantly attenuated Ang II-induced PKC phosphorylation and attenuated the Ang II-induced overexpression of NOX2 and NOX4, but not NOX1. NOX1, NOX2, NOX4, and NOX5 are reported to be regulated by Ang II in vascular cells [26], whereas NOX2 and NOX4 are mainly expressed isoforms in cardiomyocytes [27]. Consistent with our results, Zhang et al. demonstrated that Ang II produced excessive ROS via an increase in NOX2 activity, which subsequently increased calcium transients, accelerated contractility, and, finally, induced cardiac hypertrophy and remodeling [28]. Another study also showed that Ang II induced oxidative stress, hypertension, and cardiac hypertrophy in a renal artery ligation model, mainly via an increase in NOX4 protein abundance [29]. Regarding the effect of MRN on the regulation of NOX expression, our results suggested that MRN may regulate NOX expression in a NF-κB-dependent manner. A number of previous studies have demonstrated that NF-κB activation is a key event for the Ang II-induced signaling pathways, the transcriptional regulation of NOXs, and the NOX-dependent ROS production in cardiovascular cells [19,20,21].

In addition, MRN was found to regulate Ang II-induced RAGE overexpression via the attenuation of HMGB1 secretion induced by AT1R activation. HMGB1 was reported as an important mediator for RAGE activation under Ang II stimulation using anti-HMGB1 antibody treatment [22]. Kikuchi et al. showed that angiotensin receptor blockers (ARBs) inhibited the HMGB1/RAGE axis, and then suggested that ARBs may prevent stroke as well as be an effective treatment for stroke, although additional clinical studies are necessary for verification [30]. As MRN inhibited Ang II-induced AT1R/RAGE overexpression through the inhibition of HMGB1 secretion and Ang II-mediated downstream signaling, we also speculate that MRN may also be useful for the prevention of stroke as well as cardiac hypertrophy. Nevertheless, additional in vivo studies are necessary to verify the efficacy of MRN.

In addition to our results, various gradients and extracts from different parts of lotuses have shown good therapeutic effects for cardiovascular diseases. Neferine, obtained from lotus seed, was reported to have strong anti-arrhythmic potential and anti-platelet aggregation activity in rabbits [16], and to be effective in the prevention of sudden death due to myocardial ischemia-induced damage [31]. Lotus leaf extract was beneficial in the management of hyperglycemia and dyslipidemia in diabetic animal models [32]. We previously analyzed MRN and identified β-sitosterol, hyperoside, and astragalin from gradient elutions [33]. These bioactive components were reported to show anti-inflammatory and antioxidant effects in both in vitro and in vivo systems [34,35,36]. Therefore, we may presume that the antioxidant effect of MRN results from these components. However, the effects of these components on cardiovascular diseases have still not been thoroughly investigated. 

## 4. Materials and Methods 

### 4.1. Plant Material

The receptaculum of *Nelumbo nucifera* was collected in Muan, Jeonnam province, Korea (34.940720° N, 126.462064° E), in August 2011. The plant was identified by Professor Hui Kim (Mokpo National University, Muan, Korea), and a voucher specimen (JTKM-2011-01) was deposited in the herbarium at the National Development Institute of Korean Medicine.

### 4.2. Preparation of Receptaculum Extracts 

The receptaculum of *Nelumbo nucifera* (4.5 kg) was extracted three times with 100% MeOH (3 × 4 h) under reflux and filtered. The filtrate was evaporated under vacuum to yield the MeOH extract (765 g).

### 4.3. Cell Culture and Treatment

H9c2 cells (an embryonic rat cardiomyocyte cell line) were purchased from ATCC (CRL-1446, Manassas, VA, USA) and were maintained at 37 °C in a humidified atmosphere containing 5% CO2 and 95% air. The cells were cultured in Dulbecco’s modified eagle’s medium (DMEM; Gibco, USA) supplemented with 10% fetal bovine serum (FBS; Gibco) and 5% penicillin (Gibco, Waltham, MA, USA). The medium was replaced every three days. To induce myocardial hypertrophy, the cells were cultured to 70%–80% confluency, the culture medium was replaced with low serum medium (0.5% FBS) for 7 h, and then the cells were treated with Ang II (Sigma-Aldrich Corp., St. Louis, MO, USA) for 48 h. To examine the inhibitory effect of MRN on myocardial hypertrophy, MRN pretreatment was applied for 6 h before Ang II treatment.

### 4.4. Measurement of Cell Viability and LDH Assay

Cell viability was measured using the Cell Counting Kit-8 (Dojindo, Japan) assay, which is a colorimetric assay based the reduction of WST-8 by cellular dehydrogenases to produce an orange formazan product. H9c2 cells (10,000 cells/well) were seeded in a 96-well plate. After 54 h of lotus treatment, serum-free DMEM containing 10 µL Cell Counting Kit-8 (CCK-8) solution was added to each well of the 96-well plate, and then the plate was incubated for 2 h at 37 °C. The absorbance at 450 nm was measured using a microplate reader. All measurements were performed in quadruplicate. Cytotoxicity was detected by assay of the culture medium using a lactate dehydrogenase (LDH) cytotoxicity detection kit (TaKaRa Bio Inc., Kusatsu, Japan) in accordance with the manufacturer’s instructions. 

### 4.5. Cell Size Measurement

To analyze the cell surface area (CSA), H9c2 cells were stained using the F-actin stain kit (Invitrogen) in accordance with the manufacturer’s instructions. Immunofluorescence was detected via confocal microscopy (LSM780; Carl Zeiss, Germany) and the cell surface area was measured using a quantitative image analysis program (Image J, National Institutes of Health, Bethesda, MD, USA). 

### 4.6. Measurement of Total Protein Synthesis

The occurrence of Ang II-induced hypertrophy was also confirmed using Click-iT protein synthesis assay kits (Molecular Probes, USA). H9c2 cells were seeded in a 4-well chamber slide, and then treated with lotus and Ang II for 48 h. Subsequently, the drug-containing medium was removed, and 1 mL/well of medium with 50 nM Click-iT HPG working solution was added and incubated for 30 min. After incubation, the medium containing Click-iT HPG was removed and the cells were washed once with phosphate-buffered saline (PBS), which was then removed. Next, 1 mL/well 3.7 % formaldehyde in PBS was added, and the cells were incubated for 15 min at room temperature. The fixative solution was removed, the cells were washed twice with 3% bovine serum albumin (BSA) in PBS, and the cells were incubated for 20 min at room temperature with 1 mL/well 0.5% Triton X-100 in PBS. The Click-iT reaction, with Alexa-594 detection reagents, was performed for 30 min at room temperature while protected from light. After the Click-iT reaction, the protein products in cells were detected using fluorescence microscopy. 

### 4.7. ROS Detection Assay

Intracellular ROS generation was determined using CM-H2DCFDA (Molecular Probes, USA). After Ang II treatment for 30 min or 24 h, H9c2 cells were washed with PBS twice and incubated with 5 µM DCFH-DA solution in a serum-free medium at 37 °C for 20 min in the dark. Fluorescence was measured by fluorescence microscopy after the assay was performed, in accordance with the manufacturer’s instructions.

### 4.8. Immunoblotting Analysis

Proteins were extracted from the H9c2 cells and lysed with radioimmunoprecipitation assay (RIPA) buffer (Biosesang, Korea) containing a protease and phosphatase inhibitor cocktail (Thermo Scientific Inc., Waltham, MA, USA). Protein concentrations were measured using the bicinchoninic acid (BCA, Sigma) protein assay and 20 µg of the cell lysate was electrophoresed on a 10% SDS-PAGE gel. Subsequently, proteins were transferred onto polyvinylidene difluoride membranes (ATTO, Japan) and non-specific binding was blocked by incubation of the membrane in 5% non-fat milk at room temperature for 1 h. After incubation with primary antibodies specific for anti-ANP (ab209232, Abcam), anti-BNP (ab19645, Abcam), anti-NFATc1 (sc-7294, Santa Cruz), anti-phospho-PKC (#9371, Cell Signaling Technologies), anti-phospho-ERK1/2 (#4370, Cell Signaling Technologies), anti-PKC (P5704, Sigma-Aldrich), anti-ERK (#4696, Cell Signaling Technologies), anti-AT1R (AAR-011, Alomone), anti-NOX1 (ab131088, Abcam), anti-NOX2 (ab129068, Abcam), anti-NOX4 (ab109225, Abcam), anti-NF-κB (sc8088, Santa Cruz), anti-phospho-NF-κB (#3033, Cell Signaling Technologies), anti-GAPDH (sc-32233, Santa Cruz), and anti-β-actin (sc-47774, Santa Cruz) at 4 °C overnight, the membranes were incubated with HRP-conjugated secondary antibodies (Gendepot) for 1 h at room temperature. Signals were detected using enhanced chemiluminescent reagent (GE Healthcare, UK) and were quantified using ImageJ software (National Institutes of Health, USA).

### 4.9. Statistical Analysis

The data are expressed as the mean ± SEM. The significance of differences was estimated by one-way ANOVA followed by Bonferroni’s post-hoc comparison test. A *p* value of <0.05 was considered statistically significant. All data analysis was performed using the commercially available software GraphPad Prism 5.

## 5. Conclusions

Collectively, our results have demonstrated the potential of the extract of lotus receptacles as a preventive or therapeutic drug, as MRN was effective in the amelioration of in vitro Ang II-induced cardiac hypertrophy via the regulation of AT1R-RAGE overexpression and excessive oxidative stress. These results may support the use of lotus as a functional food. 

## Figures and Tables

**Figure 1 molecules-24-01647-f001:**
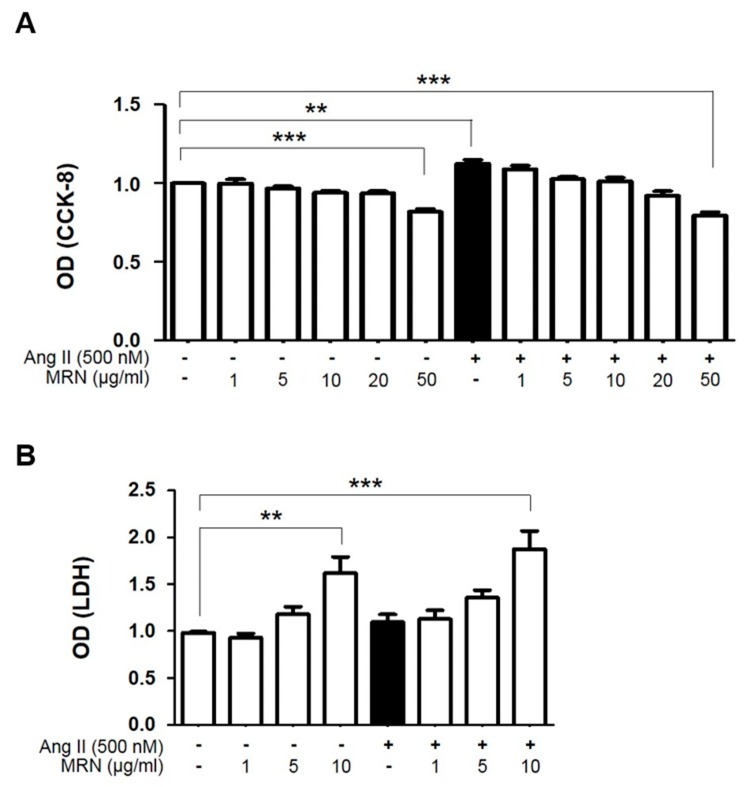
Effects of *Nelumbo nucifera* receptaculum extract on H9c2 cell viability. (**A**) Cell viability was measured by using the Cell Counting Kit-8 (CCK-8) assay, and (**B**) cytotoxicity was measured using the LDH assay. Cell viability (CCK-8) and cytotoxicity (LDH) were measured after treatment with 1–50 µg/mL MeOH extract of receptaculum Nelumbinis (MRN) for 48 h in the presence or absence of angiotensin II (Ang II). Quantitative data were expressed as the mean ± SEM. of 3 independent experiments. ** *p* < 0.01, *** *p* < 0.001.

**Figure 2 molecules-24-01647-f002:**
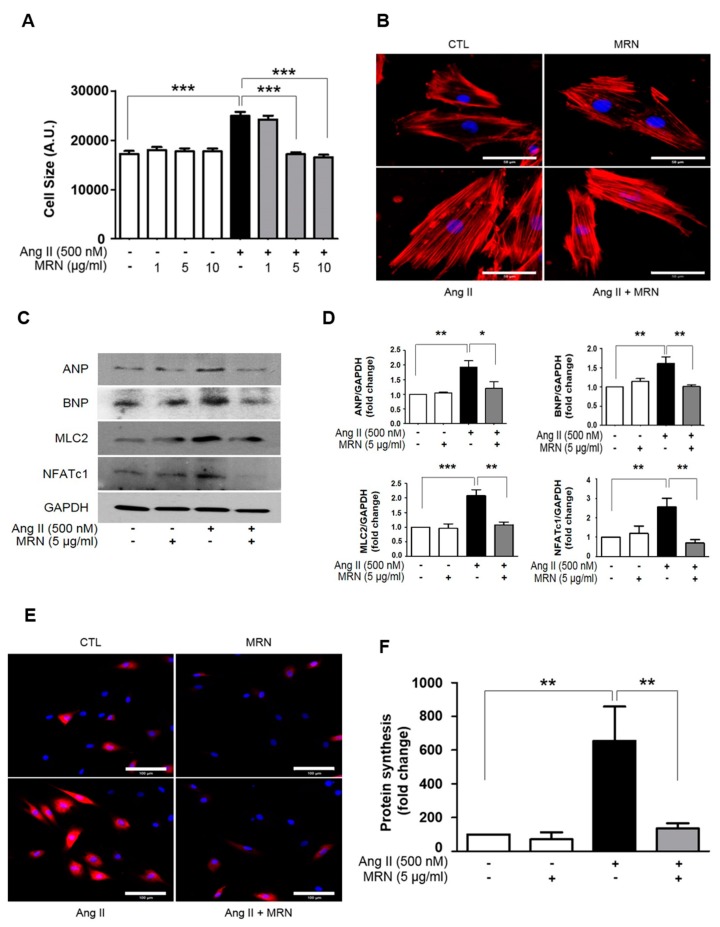
*Nelumbo nucifera* receptaculum extract (MRN) reduces Ang II-induced cardiomyocyte hypertrophy. H9c2 cells were treated with 500 nM Ang II for 48 h in the presence or absence of 5 µg/mL MRN. (**A**) Cell size was measured from 50–150 cells in each group and analyzed using ImageJ software. Results are representative of three separate experiments. *** *p* < 0.001. (**B**) F-actin in H9c2 cells was stained with Texas Red-X phalloidin and detected using confocal microscopy. Scale bar = 50 µm. (**C,D**) Western blot of expression of cardiomyocyte hypertrophic markers (ANP, BNP, NFATc1, MLC2v) and GAPDH. Results are representative of five separate experiments. * *p* < 0.05, ** *p* < 0.01, *** *p* < 0.001. (**E,F**) Protein synthesis was detected using a Click-iT HPG Alexa Fluor protein synthesis assay system. Scale bar = 100 µm. Results are representative of three separate experiments. ** *p* < 0.01. The bars represent the mean ± SEM.

**Figure 3 molecules-24-01647-f003:**
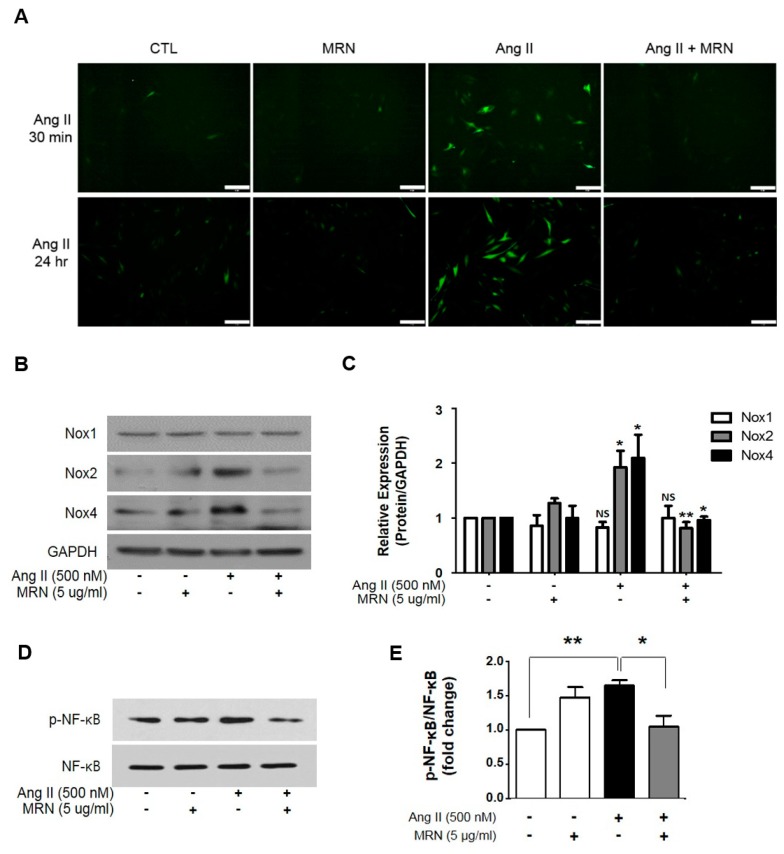
*Nelumbo nucifera* receptaculum extract (MRN) inhibits intracellular reactive oxygen species (ROS) by NADPH oxidases (NOXs) in Ang II-induced cardiomyocyte hypertrophy. H9c2 cells were treated with 500 nM Ang II for 30 min or 24 h in the presence or absence of MRN. (**A**) The effect of MRN on intracellular ROS levels was measured using the CM-DCFDA assay and fluorescence microscopy. Scale bar = 50 µm. Results are representative of three separate experiments. (**B,C**) H9c2 cells were treated with 500 nM Ang II for 24 h in the presence or absence of MRN, and the protein expression of NOX1, NOX2, NOX4, and GAPDH was examined by western blotting. Results are representative of four separate experiments. * *p* < 0.05 vs. DMSO-treated control, ** *p* < 0.01 vs. DMSO-treated control. (**D,E**) H9c2 cells were treated with 500 nM Ang II for 24 h in the presence or absence of MRN, and expression of NF-κB and p-NF-κB was examined by western blotting. Results are representative of four separate experiments. * *p* < 0.05, ** *p* < 0.01. The bars represent the mean ± SEM.

**Figure 4 molecules-24-01647-f004:**
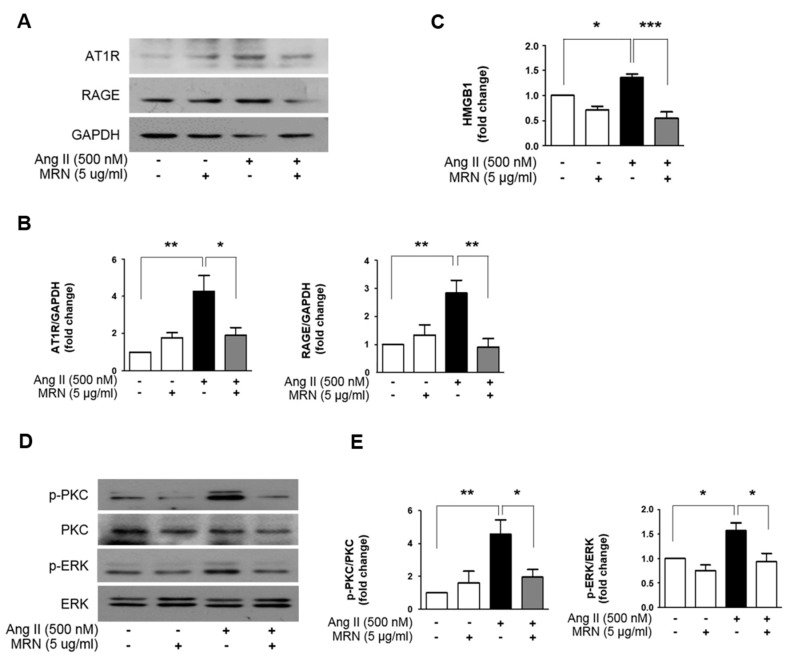
*Nelumbo nucifera* receptaculum extract (MRN) reduces protein kinase C (PKC)– extracellular-signal-regulated kinase (ERK) activation in Ang II-induced cardiomyocyte hypertrophy. H9c2 cells were treated with 500 nM Ang II for 24 h in the presence or absence of MRN. (**A**,**B**) Western blots of type I angiotensin receptor (AT1R) and RAGE expression are shown. Results are representative of five separate experiments. * *p* < 0.05, ** *p* < 0.01, *** *p* < 0.001. (**C**) The supernatant was collected and the release of HMGB1 was measured by ELISA. Results are representative of six separate experiments. * *p* < 0.05, ** *p* < 0.01. (**D**,**E**) H9c2 cells were treated with 500 nM Ang II in the presence or absence of MRN for 20 min and analyzed by western blotting. The values of the relative p-PKC and p-ERK bands were normalized to the PKC and ERK bands to represent the relative abundance of the p-PKC and p-ERK protein, respectively. Results are representative of four separate experiments. * *p* < 0.05, ** *p* < 0.01. The bars represent the mean ± SEM.

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
