# Peer review of "Nelumbo nucifera Receptaculum Extract Suppresses Angiotensin II-Induced Cardiomyocyte Hypertrophy"

_molecules, 2019, doi:10.3390/molecules24091647_

Round 1

Reviewer 1 Report

The quality of Western Blots (Figure 2 and 3) is not good and should be repeated for concrete conclusion. The bands show that either transfer or detection was poor. Sample size is already very small (n=3) and authors are showing one representative blot (which are of poor quality) of which they performed densitometry analysis. If the quality of blots is poor, densitometry analysis cannot be considered conclusive.

Author Response

response attached as a seperate file.

Reviewer 2 Report

Dear Authors

This methology is perfect and your results are interesting but you must improve:

line 205 - GPS coordinates should be added,

Figure 1 - unfortunately unreadable

Figure 2 - please mark representative areas

Figure 3 - please mark representative areas

All literature should contain DOI

Author Response

We authors very much appreciated the encouraging, critical and constructive comments and suggestions on this manuscript by the reviewers. The comments have been very thorough and useful in improving the manuscript. We strongly believe that the comments and suggestions have significantly increased the scientific value of revised manuscript. We are submitting the corrected manuscript with consolidated data. The manuscript has been revised as per the comments given by the reviewer, and our responses to all the comments are as follows:

 Comments and Suggestions for Authors

This methodology is perfect and your results are interesting but you must improve:

1. line 205 - GPS coordinates should be added.

Response: The GPS coordinates for the Muan, Jeonnam province of Korea has been added in the revised manuscript as the reviewer requested.

2. Figure 1 - unfortunately unreadable

Response: The Figure 1 has been replaced by a better one. Sorry for the inconvenience.

3. Figure 2 - please mark representative areas

Response: First of all, the comment was a bit ambiguous since there are 6 items in the Figure 2. However, we assumed that the comment was for the Figure 2B and 2E. If it was indeed for the Figure 2B and 2E, there must have been a misunderstanding. The images of Figure 2B and 2E themselves were chosen as representative images of each group. For example, in the Figure 2B, Ang II-treated cells have a significantly larger cell size compared to that of CTL or MRN treated group. In these images, the F-actin staining was simply to show the outline of cells. In other words, marking representative area is neither necessary nor possible for the image in Figure 2B. However, if we authors misunderstood the comment, please let us know.

4. Figure 3 - please mark representative areas

Response: The same principle explained above also applies to the Figure 3A. In the Figure 3A, the green fluorescence indicates intracellular ROS generated. Therefore, the each image in the Figure 3A already represents the general cellular situation of the group they belong to (i.e., compared to control group, a lot more cells were generating ROS after Ang II treatment in layman’s terms). Therefore, we authors believe that it is not necessary to mark representative area for those images. However, if we authors misunderstood the comment, please let us know.

5. All literature should contain DOI

Response: DOI has been added in the revised manuscript as the reviewer requested.

Reviewer 3 Report

In this manuscript the authors convincingly show that a methanolic extract of Lotus receptaculum powder (MRN) inhibits angiotensin II (AngII)-induced cardiomyocyte hypertrophic responses and attenuates AngII-induced NOX2 and NOX4 expression in the rat cell line Hgc2. The authors conclude that the MRN extract exerts its anti-hypertrophic effect through the regulation of intracellular ROS homeostasis and the ROS-mediated signaling through the PKC-ERK pathway. The paper is very clearly written and the experiments follow a logical reasoning.

Major: It is not clear from the presented data that MRN prevents AngII-induced hypertrophic responses by its antioxidant activity on intracellularly produced ROS  (abstract, lines 28-30).

From Figure 3 the authors conclude that MRN inhibits intracellular ROS (levels ?) by NOXs, but this conclusion is not supported by the data. Figure 3 only shows prevention of an increase in intracellular ROS levels early after stimulation with AngII, and late increase in NOXs protein level. In addition, there is no evidence presented that the increase in ROS levels is due to NOX activity, for example by using NOX inhibitors. Further, MRN prevents AngII-induced PKC phosphorylation within 20 minutes, suggesting that MRN may interfere primarily with early AngII-AT1R signal transduction and only secondary by ROS-induced cardiomyocyte damage. Are the MRN effects be mimicked with a pure anti-oxidant as a positive control?

Minor:  

Figure 1B suggests that cell toxicity is increased already with 5 ug/ml MRN, considering the small SEM. What is the number of independent replicates here? Why is this cytotoxic effect of 5ug/ml MRN ignored in designing the subsequent experiments?

It is unlikely that the scale bars in figure 2B and 2E are identical (50 um)?

Line 22: please replace ‘myocardial hypertrophy’ by ‘cardiomyocyte hypertrophy’.

Line 139: MLC2 is a hypertrophy marker, not GAPDH

Lines 140 and 142: p<0.001 is not shown in figures 2C-D or 2E-F

Line 151: please define control

Author Response

We authors very much appreciated the encouraging, critical and constructive comments and suggestions on this manuscript by the reviewers. The comments have been very thorough and useful in improving the manuscript. We strongly believe that the comments and suggestions have significantly increased the scientific value of revised manuscript. We are submitting the corrected manuscript with consolidated data. The manuscript has been revised as per the comments given by the reviewer, and our responses to all the comments are as follows:

Comments and Suggestions for Authors 3

In this manuscript the authors convincingly show that a methanolic extract of Lotus receptaculum powder (MRN) inhibits angiotensin II (AngII)-induced cardiomyocyte hypertrophic responses and attenuates AngII-induced NOX2 and NOX4 expression in the rat cell line H9c2. The authors conclude that the MRN extract exerts its anti-hypertrophic effect through the regulation of intracellular ROS homeostasis and the ROS-mediated signaling through the PKC-ERK pathway. The paper is very clearly written and the experiments follow a logical reasoning.

 Major: It is not clear from the presented data that MRN prevents AngII-induced hypertrophic responses by its antioxidant activity on intracellularly produced ROS (abstract, lines 28-30). From Figure 3 the authors conclude that MRN inhibits intracellular ROS (levels ?) by NOXs, but this conclusion is not supported by the data. Figure 3 only shows prevention of an increase in intracellular ROS levels early after stimulation with AngII, and late increase in NOXs protein level. In addition, there is no evidence presented that the increase in ROS levels is due to NOX activity, for example by using NOX inhibitors. Further, MRN prevents AngII-induced PKC phosphorylation within 20 minutes, suggesting that MRN may interfere primarily with early AngII-AT1R signal transduction and only secondary by ROS-induced cardiomyocyte damage. Are the MRN effects be mimicked with a pure anti-oxidant as a positive control?

Response: First, as to the comment that “Figure 3 only shows prevention of an increase in intracellular ROS levels early after stimulation with Ang II, and late increase in NOXs protein level”, our data shows that 24 hours of Ang II stimulation increased both intracellular ROS production (Figure 3A, bottom panel) and the expression of NOX-2 and -4 (Figure 3B). We authors are not sure whether the “early” and “late” mentioned by the reviewer refer to “30 min” and “24hr”, respectively. However, it is clear that, at least 24 hrs after the Ang II treatment, both intracellular ROS and NOX expression increased, and such Ang II-induced increase was abrogated by MRN treatment.

For the comment that there was no evidence presented that the increase in ROS levels is due to NOX activity, it is correct that we did not provide empirical data using NOX inhibitors. However, it is well-established fact that Ang II induces ROS production in a NOX-dependent manner. To be more specific, AT1R initially generates intracellular ROS in both PKC and NOX-dependent manner. The increased ROS, such as hydrogen peroxide, impacts the mitochondrial matrix stimulating mitochondrial ROS production. Again the increase mitochondrial ROS can exit the mitochondria and provide the feed-forward stimulation of cytoplasmic NOX [1,2]. Furthermore, since the major interest of the present study was to examine the effect of MRN on Ang II-induced hypertrophy rather than to investigate the role of NOX in Ang II-induced ROS production, we authors do not believe that proving the Ang II-induced ROS production in this study was in fact NOX-dependent using NOX inhibitor is absolutely necessary for the present study. Nevertheless, as alternative, the effect of MRN on the Ang II-induced NF-κB activation was examined because the activation of NF-κB is a key event for the Ang II-induced signaling pathways [3,4], the transcriptional regulation of NOXs and the NOX-dependent ROS production as well [5]. The newly added data indicated that MRN significantly attenuated the Ang II-induced NF-κB activation and is presented as Figure 3D in the revised manuscript.

1.             Dikalov, S.I.; Nazarewicz, R.R. Angiotensin II-induced production of mitochondrial reactive oxygen species: potential mechanisms and relevance for cardiovascular disease. Antioxid Redox Signal 2013, 19, 1085-1094, doi:10.1089/ars.2012.4604.

2.             Hingtgen, S.D.; Tian, X.; Yang, J.; Dunlay, S.M.; Peek, A.S.; Wu, Y.; Sharma, R.V.; Engelhardt, J.F.; Davisson, R.L. Nox2-containing NADPH oxidase and Akt activation play a key role in angiotensin II-induced cardiomyocyte hypertrophy. Physiol Genomics 2006, 26, 180-191, doi:10.1152/physiolgenomics.00029.2005.

3.             Purcell, N.H.; Tang, G.; Yu, C.; Mercurio, F.; DiDonato, J.A.; Lin, A. Activation of NF-kappa B is required for hypertrophic growth of primary rat neonatal ventricular cardiomyocytes. Proc Natl Acad Sci U S A 2001, 98, 6668-6673, doi:10.1073/pnas.111155798.

4.             Freund, C.; Schmidt-Ullrich, R.; Baurand, A.; Dunger, S.; Schneider, W.; Loser, P.; El-Jamali, A.; Dietz, R.; Scheidereit, C.; Bergmann, M.W. Requirement of nuclear factor-kappaB in angiotensin II- and isoproterenol-induced cardiac hypertrophy in vivo. Circulation 2005, 111, 2319-2325, doi:10.1161/01.CIR.0000164237.58200.5A.

5.             Manea, A.; Tanase, L.I.; Raicu, M.; Simionescu, M. Transcriptional regulation of NADPH oxidase isoforms, Nox1 and Nox4, by nuclear factor-kappaB in human aortic smooth muscle cells. Biochem Biophys Res Commun 2010, 396, 901-907, doi:10.1016/j.bbrc.2010.05.019.

Minor: 

1. Figure 1B suggests that cell toxicity is increased already with 5 ug/ml MRN, considering the small SEM. What is the number of independent replicates here? Why is this cytotoxic effect of 5ug/ml MRN ignored in designing the subsequent experiments?

Response: A total of 6 independent replicates were used. However, additional experiments were performed to increase the reliability and 6 more replicates were added in the revised manuscript. After increasing the number of replicates, 5 ug/ml of MRN increased LDH activity by approximately 1.2-fold compared to untreated control without statistical significance. For deciding a proper concentration of experiments in our lab, there are three main parameters used; CCK data, LDH data, and morphological examination. As the reviewer pointed out, initially, 5 ug/ml MRN showed about 1.5-fold increase in cytotoxicity (LDH assay). The LDH assay basically detects the extracellular leakage of LDH due to compromised membrane integrity. However, the data obtained by LDH test are not necessarily always proportional to the actual cell death occurred because even repairable, temporary membrane damages can induce LDH release. For example, the common electroporation technic widely used to introduce exogenous nucleic acids, which does not cause significant cell death, can also increase LDH release (World J Gastroenterol 2002;8(5):893-896). Furthermore, CCK data showed that 5 ug/ml MRN had no significant effect on the cell viability, and there was no obvious morphological changes indicating cell death observed. Based on these observations, we have decided the concentration of MRN for further studies.

2. It is unlikely that the scale bars in figure 2B and 2E are identical (50 um)?

Response: The reviewer is correct. The scale bar in Figure 2B and Figure 2E are 50 µm and 100 µm, respectively. Necessary correction has been made in the revised manuscript.

3. Line 22: please replace ‘myocardial hypertrophy’ by ‘cardiomyocyte hypertrophy’.

Response: The term ‘myocardial hypertrophy’ has been replaced with ‘cardiomyocyte hypertrophy’ as the reviewer recommended.

4. Line 139: MLC2 is a hypertrophy marker, not GAPDH

Response:  Thank you for letting us know this mistake. This one was corrected in the manuscript.

5. Lines 140 and 142: p<0.001 is not shown in figures 2C-D or 2E-F

Response: indication of significance has been corrected in the revised manuscript.

6. Line 151: please define control

Response: The control is a DMSO-treated group and this definition has been added to the figure legend for figures 3 of the revised manuscript.   

Reviewer 4 Report

This is a nice study addressing the potential therapeutic effect of receptacles of lotus on pathological cardiomyocyte hypertrophy. The study is well designed using a known experimental protocol.

I woul dlike to suggest authors to use TEMPOL and DPI for comparison of intracellular ROS production and inhibition by MRN. TEMPOL is an intracellular antioxidant while DPI is an inhibitor of NADPH Oxidase in order tho strengthened their findings.

Moreover, it would be nice to speculate any possible use in clinical pratice indicating possible dosage if available 

Author Response

Comments and Suggestions for Authors

This is a nice study addressing the potential therapeutic effect of receptacles of lotus on pathological cardiomyocyte hypertrophy. The study is well designed using a known experimental protocol.

I would like to suggest authors to use TEMPOL and DPI for comparison of intracellular ROS production and inhibition by MRN. TEMPOL is an intracellular antioxidant while DPI is an inhibitor of NADPH Oxidase in order to strengthen their findings.

Moreover, it would be nice to speculate any possible use in clinical practice indicating possible dosage if available 

Response: Thank you for your valuable suggestion. However, it is well-established fact that Ang II induces ROS production in a NOX-dependent manner. To be more specific, AT1R initially generates intracellular ROS in both PKC and NOX-dependent manner. The increased ROS, such as hydrogen peroxide, impacts the mitochondrial matrix stimulating mitochondrial ROS production. Again the increase mitochondrial ROS can exit the mitochondria and provide the feed-forward stimulation of cytoplasmic NOX [1,2]. Furthermore, since the major interest of the present study was to examine the effect of MRN on Ang II-induced hypertrophy rather than to investigate the role of NOX in Ang II-induced ROS production, we authors do not believe that proving the Ang II-induced ROS production in this study was in fact NOX-dependent using NOX inhibitor is absolutely necessary for the present study. Nevertheless, as alternative, the effect of MRN on the Ang II-induced NF-κB activation was examined because the activation of NF-κB is a key event for the Ang II-induced signaling pathways [3,4], the transcriptional regulation of NOXs and the NOX-dependent ROS production as well [5]. The newly added data indicated that MRN significantly attenuated the Ang II-induced NF-κB activation and is presented as Figure 3D in the revised manuscript.

Regarding any possible use in clinical practice indicating possible dosage, we authors could not speculate about clinical practice at this stage because MRN is a mixture of known and unknown substance. Instead of that, we added description on the clinical effect of a number of compounds in the end of Discussion section. So we agree that further study using MRN-derived single compound might be needed in the near future.

1.             Dikalov, S.I.; Nazarewicz, R.R. Angiotensin II-induced production of mitochondrial reactive oxygen species: potential mechanisms and relevance for cardiovascular disease. Antioxid Redox Signal 2013, 19, 1085-1094, doi:10.1089/ars.2012.4604.

2.             Hingtgen, S.D.; Tian, X.; Yang, J.; Dunlay, S.M.; Peek, A.S.; Wu, Y.; Sharma, R.V.; Engelhardt, J.F.; Davisson, R.L. Nox2-containing NADPH oxidase and Akt activation play a key role in angiotensin II-induced cardiomyocyte hypertrophy. Physiol Genomics 2006, 26, 180-191, doi:10.1152/physiolgenomics.00029.2005.

3.             Purcell, N.H.; Tang, G.; Yu, C.; Mercurio, F.; DiDonato, J.A.; Lin, A. Activation of NF-kappa B is required for hypertrophic growth of primary rat neonatal ventricular cardiomyocytes. Proc Natl Acad Sci U S A 2001, 98, 6668-6673, doi:10.1073/pnas.111155798.

4.             Freund, C.; Schmidt-Ullrich, R.; Baurand, A.; Dunger, S.; Schneider, W.; Loser, P.; El-Jamali, A.; Dietz, R.; Scheidereit, C.; Bergmann, M.W. Requirement of nuclear factor-kappaB in angiotensin II- and isoproterenol-induced cardiac hypertrophy in vivo. Circulation 2005, 111, 2319-2325, doi:10.1161/01.CIR.0000164237.58200.5A.

5.             Manea, A.; Tanase, L.I.; Raicu, M.; Simionescu, M. Transcriptional regulation of NADPH oxidase isoforms, Nox1 and Nox4, by nuclear factor-kappaB in human aortic smooth muscle cells. Biochem Biophys Res Commun 2010, 396, 901-907, doi:10.1016/j.bbrc.2010.05.019.

Round 2

Reviewer 1 Report

Authors addressed the issues and improved the quality of data.

Author Response

We attached a certificate of English editing for this manuscript.

Reviewer 3 Report

I thank the authors for their extensive response to my concerns about the last sentence of the abstract. Unfortunately, they have not convinced me. My problem remains the early (<30 min) AngII induced changes in PKC and ERK activation (fig 4DE), and the initial increase in ROS production (fig. 3A). My interpretation of these findings is that MRN affects primarily AngII-induced signaling to PKC and ERK, which subsequently  leads to reduced ROS production, and not necessarily the other way around. In this case MRN is not acting primarilly through its alleged anti-oxidant properties. As the authors explain, ROS leads to extra ROS through a feed-forward on NOX. If MRN prevents early ROS production secondary to prevention of angII induced PKC and ERK activation, it will subsequently also lead to reduced feed forward regulation of NOX, and to prevention of hypertrophy. Measuring NOX protein levels or pNF-KB levels only after 24 h therefore does not help to settle this dispute. So, in my mind the conclusion that ...MRN exerted a significant protective effect against cardiomyocyte hypertrophy through the regulation of intracellular ROS homeostasis and the ROS-mediated PKC-ERK signaling axis... is not supported by the presented data.

Author Response

Dear Editor,

We authors very much appreciated the encouraging, critical and constructive comments and suggestions on this manuscript by the reviewers. We believe that the comments and suggestions have significantly increased the scientific value of revised manuscript. We are re-submitting the corrected manuscript. The manuscript has been revised as per the comments given by the reviewer, and our response to the comment is as follows:

Author's Reply to the Review Report (Reviewer 3)

I thank the authors for their extensive response to my concerns about the last sentence of the abstract. Unfortunately, they have not convinced me. My problem remains the early (<30 min) AngII induced changes in PKC and ERK activation (fig 4D-E), and the initial increase in ROS production (fig. 3A). My interpretation of these findings is that MRN affects primarily AngII-induced signaling to PKC and ERK, which subsequently  leads to reduced ROS production, and not necessarily the other way around. In this case MRN is not acting primarilly through its alleged anti-oxidant properties. As the authors explain, ROS leads to extra ROS through a feed-forward on NOX. If MRN prevents early ROS production secondary to prevention of angII induced PKC and ERK activation, it will subsequently also lead to reduced feed forward regulation of NOX, and to prevention of hypertrophy. Measuring NOX protein levels or pNF-KB levels only after 24 h therefore does not help to settle this dispute. So, in my mind the conclusion that ...MRN exerted a significant protective effect against cardiomyocyte hypertrophy through the regulation of intracellular ROS homeostasis and the ROS-mediated PKC-ERK signaling axis... is not supported by the presented data.

Response: The reviewer’s criticism is correct. The sentence in issue should have been revised during the last round of revision, but it had not been. The phraseROS-mediated PKC-ERK signaling” is incorrect since the MRN-mediated PKC-ERK signaling precedes the resultant downregulation of AngII-induced ROS production. Accordingly, the sentence has been changed to MRN exerted a significant protective effect against the Ang II-induced cardiomyocyte hypertrophy through suppression of the PKC-ERK signaling which subsequently lead to attenuation of the intracellular ROS production” in the revised manuscript. We authors thank you for your valuable criticism.